# The Role of Pre-Treatment Inflammatory Biomarkers in Predicting Tumor Response to Neoadjuvant Chemoradiotherapy in Rectal Cancer

**DOI:** 10.3390/medicina61050865

**Published:** 2025-05-08

**Authors:** Yunus Emre Altıntaş, Ahmet Bilici, Özcan Yıldız, Oğuzcan Kınıkoğlu, Ömer Fatih Ölmez

**Affiliations:** Department of Medical Oncology, Faculty of Medicine, Istanbul Medipol University, 34000 Istanbul, Türkiye; abilici@medipol.edu.tr (A.B.); oyildiz@medipol.edu.tr (Ö.Y.); ogokinikoglu@yahoo.com (O.K.); ofolmez@medipol.edu.tr (Ö.F.Ö.)

**Keywords:** rectal cancer, neoadjuvant chemoradiotherapy, inflammatory markers, neutrophil-to-lymphocyte ratio

## Abstract

*Background and Objectives*: This study aimed to investigate the predictive and prognostic value of pre-treatment systemic inflammatory markers in patients with locally advanced rectal cancer (RC) undergoing neoadjuvant chemoradiotherapy (CRT) or radiotherapy (RT) alone. *Materials and Methods*: A total of 79 patients with biopsy-confirmed locally advanced RC treated at a single tertiary center between 2011 and 2017 were retrospectively analyzed. Pre-treatment blood-based inflammatory indices, including neutrophil-to-lymphocyte ratio (NLR), derived NLR, platelet-to-lymphocyte ratio (PLR), lymphocyte-to-monocyte ratio (LMR), and hemoglobin levels, were recorded. Tumor response was assessed using the Ryan tumor regression grade (TRG), and associations between laboratory parameters, treatment response, and recurrence-free survival (RFS) were evaluated. *Results*: Among 79 patients (mean age: 55.9 ± 11.98 years; 67.1% male), 57 received neoadjuvant CRT and 22 underwent short-course RT. Complete pathological response (pCR) was observed in 10 patients (12.7%). No statistically significant associations were found between baseline inflammatory markers and TRG, tumor differentiation, or pCR. ROC analysis revealed that none of the markers demonstrated significant discriminatory power for predicting tumor response or recurrence. However, a weak but statistically significant inverse correlation was identified between poor TRG response and higher baseline values of NLR, derived NLR, and PLR (*p* < 0.05). *Conclusions*: Inflammatory biomarkers such as NLR, PLR, and LMR, while easily accessible and cost-effective, did not demonstrate strong predictive or prognostic value in this cohort of RC patients receiving neoadjuvant therapy. These findings suggest that reliance solely on systemic inflammatory indices may be insufficient for predicting treatment outcomes, emphasizing the need for integrative models incorporating molecular and pathological markers.

## 1. Introduction

Colorectal cancer (CRC) ranks as the fourth most common cancer and is the second leading cause of cancer-related mortality in the United States [1]. Malignancies located within 15 cm from the anal verge are classified as rectal cancer (RC), accounting for approximately 30% of CRC, with the remaining 70% being colon cancer. Treatment options include surgery, neoadjuvant/adjuvant chemoradiotherapy (CRT), and palliative chemotherapy. Surgical resection remains the cornerstone of curative treatment for RC. However, surgery alone provides high cure rates only for early-stage patients. Five-year survival rates following curative resection range from 80 to 90% for stage I disease but decline to approximately 70% for stage II and III disease [2,3].

With the advancement of systemic treatment strategies such as neoadjuvant CRT, five-year survival rates have improved in recent years [2,3]. Neoadjuvant CRT has been established as the standard treatment strategy, particularly for locally advanced RC [4]. The introduction of neoadjuvant CRT has significantly improved survival outcomes for patients with locally advanced RC [5,6]. However, patients who do not achieve a complete pathological response to CRT tend to experience worse outcomes and are at a higher risk of local or metastatic recurrence at any time.

The five-year disease-free survival (DFS) rate is reported to be 83% in patients achieving a pathological response, whereas it drops to 66% in non-responders [7]. Therefore, it remains crucial to identify the patients who would benefit the most from CRT and to discover specific biomarkers that can help predict treatment response, ultimately leading to better patient selection. Such an approach could prevent unnecessary radiation therapy and surgical interventions [8,9]. Consequently, the identification of appropriate biomarkers has become an important area of research [10]. Several studies have investigated the role of systemic inflammatory response in tumor behavior and prognosis. Elevated C-reactive protein (CRP), an acute-phase reactant, has been associated with poor outcomes in multiple solid tumors, including CRC.

Several studies have demonstrated that pre-treatment inflammatory markers obtained from routine blood tests, such as the NLR, PLR, and C-reactive protein-to-albumin ratios, are associated with both short- and long-term outcomes in colorectal cancer. Elevated NLR and PLR have been linked to higher rates of postoperative complications and worse disease-free and overall survival, while increased CAR is associated with poor survival in rectal cancer patients receiving neoadjuvant therapy [11].

During an acute inflammatory response, alterations occur in different leukocyte subsets. A relative lymphopenia is often accompanied by an increased neutrophil count. Therefore, calculating the NLR provides a simple measure of systemic inflammation [12,13].

In this study, we aimed to evaluate the relationship between pre-treatment serum biomarkers, including hemoglobin, lymphocyte, neutrophil, and NLR, and treatment response criteria, as well as TRG observed in histopathological examination, in patients diagnosed with clinically advanced rectal carcinoma who received neoadjuvant CRT or radiotherapy alone. Additionally, we aimed to assess the impact of these biomarkers on survival outcomes and to identify potential predictive factors for treatment response.

## 2. Materials and Methods

### 2.1. Study Population and Variables

The clinical records of 79 patients with locally advanced RC who underwent surgery at the Department of Internal Medicine, Division of Oncology, Medipol University Faculty of Medicine, between 2011 and 2017 were retrospectively analyzed from a prospectively maintained database. This study was approved by the Medipol University Non-Interventional Clinical Research Ethics Committee (Approval No: 10840098-604.01.01-20058). All patients had a biopsy-confirmed diagnosis of locally advanced RC. Data on demographics, neoadjuvant treatment, surgical procedures, preoperative laboratory findings, clinicopathological variables, and recurrence-free survival were collected.

Tumor response to neoadjuvant therapy was assessed using the Ryan TRG system, classifying tumors as TRG1 (minimal/no residual tumor), TRG2 (fibrosis > tumor), or TRG3 (tumor > fibrosis) [14]. Pre-treatment blood count and biochemical analyses were evaluated, including neutrophil-to-lymphocyte ratio (NLR), derived NLR, lymphocyte count, hemoglobin, platelet-to-lymphocyte ratio (PLR), lymphocyte-to-monocyte ratio (LMR), and neutrophil count. The primary endpoint of this study was to assess the association between pre-treatment inflammatory biomarkers and tumor response to neoadjuvant treatment, as evaluated by pCR and TRG. Secondary endpoints included the relationship between these biomarkers and tumor differentiation, as well as recurrence-free survival (RFS). Comparisons were made between patient subgroups stratified by pathological response, TRG classification, and tumor differentiation grade. Additionally, age, pre-treatment clinical TNM stage, postoperative pathological TNM stage, and histopathological grade were recorded.

The study population was selected based on predefined inclusion and exclusion criteria. The inclusion criteria were as follows:Patients aged 18 years or older;Histologically confirmed diagnosis of locally advanced rectal adenocarcinoma;Tumor located within 15 cm of the anal verge;Undergoing neoadjuvant chemoradiotherapy (CRT) or short-course radiotherapy (RT) followed by surgical resection;Availability of complete pre-treatment laboratory data, including inflammatory markersAvailability of recurrence-free survival and pathological outcome data.

Exclusion criteria included:Presence of synchronous malignancies or history of other primary malignancies;Presence of distant metastases at the time of diagnosis (stage IV disease);Incomplete treatment data or lack of follow-up information;Non-adenocarcinoma histologies (e.g., squamous cell carcinoma, neuroendocrine tumors);Patients who did not undergo surgical resection following neoadjuvant therapy.

Tumors were classified based on location as lower (≤5 cm from the anal verge), middle (5–10 cm), or upper rectum (>10 cm). Neoadjuvant chemoradiotherapy (CRT) or short-course radiotherapy (RT) was administered, followed by low anterior resection (LAR). The stage was based on the TNM system (AJCC/UICC). Adjuvant chemotherapy was considered based on clinical status and patient preference.

In the standard neoadjuvant treatment (SNT) approach, patients first received CRT, which included either capecitabine (2 × 825 mg/m^2^) or infusional 5-fluorouracil (5-FU, 225 mg/m^2^) administered concurrently with radiotherapy. Radiotherapy was delivered either as a short-course regimen (25 Gy in 5 fractions) or a long-course regimen (45–50 Gy in 25–28 fractions). Following the completion of CRT, patients underwent surgery after a median waiting period of 8 weeks. Postoperatively, patients were administered adjuvant chemotherapy for 12–16 weeks, consisting of either FOLFOX (leucovorin 400 mg/m^2^, 5-FU 400 mg/m^2^ bolus plus 1200 mg/m^2^/day continuous infusion, oxaliplatin 85 mg/m^2^) or CAPEOX (oxaliplatin 130 mg/m^2^, capecitabine 1000 mg/m^2^) [15].

### 2.2. Statistics

The statistical analysis of the research data was performed using Statistical Package for Social Sciences (SPSS) Windows version 15 (SPSS Inc., Chicago, IL, USA). Descriptive statistics were presented as frequencies and percentages for categorical variables, while mean, median, minimum, maximum, and standard deviation (SD) values were reported for numerical variables. Homogeneity was assessed using Levene’s test, and a *p*-value > 0.05 was considered indicative of homogeneity. The distribution of continuous variables was evaluated using the Kolmogorov–Smirnov and Shapiro–Wilk normality tests, with a *p*-value > 0.05 considered indicative of normal distribution.

For comparisons between two independent groups, the independent T-test was used when the normality assumption was met, whereas the Mann–Whitney U test was applied when this assumption was not satisfied. For comparisons among three independent groups, the ANOVA test was employed if the normality assumption was met, while the Kruskal–Wallis test was used if the assumption was violated. Pearson’s chi-square (χ^2^) test or Fisher’s exact test was conducted for categorical variables. Correlation analysis was performed using Pearson’s test for parametric values and Spearman’s test for non-parametric values. The Receiver Operating Characteristic (ROC) analysis was used to determine cutoff values, sensitivity, and specificity. A *p*-value < 0.05 was considered statistically significant for all analyses.

## 3. Results

A total of 79 patients with locally advanced RC were included in this study. The mean age was 55.94 ± 11.98 years, with 53 (67.1%) male and 26 (32.9%) female patients. Pre-treatment clinical tumor staging classified 48 patients (60.8%) as stage III, 16 (20.3%) as stage II, nine (11.4%) as stage I, and six (7.6%) as stage IV (Table 1).

Among the 65 patients, according to the Ryan regression grade, 81.5% (53) were classified as TRG1 (no tumor or very little tumor with fibrosis throughout the entire wall), 6.2% (4) as TRG2 (fibrosis more predominant than the tumor), and 12.3% (8) as TRG3 (fibrosis less predominant than the tumor).

As a response to treatment, TRG was classified as G2 in 29 patients (36.7%), G1 in 24 patients (30.4%), G3 in 16 patients (20.3%), and G0 in 10 patients (12.7%).

The distance of the tumor from the anal sphincter was categorized into three groups based on centimeters (cm): 1–4 cm, 5–10 cm, and 11–15 cm. A total of 32 patients (40.5%) had tumors located 1–4 cm away, 38 patients (48.1%) had tumors 5–10 cm away, and nine patients (11.4%) had tumors 11–15 cm away.

In the postoperative pathological evaluation, vascular invasion was not detected in 63 patients (79.7%) across the entire cohort, while perineural invasion was absent in 62 patients (78.5%). Considering the radiotherapy techniques applied, 57 patients (72.2%) received long-course chemoradiotherapy, whereas 22 patients (27.8%) underwent short-course single radiotherapy. These findings are summarized in Table 2.

At the time of diagnosis, the median lymphocyte count was 1390 (range: 320–4090), while the median neutrophil count was 4520 (range: 1940–12,050). The median values for hematological indices were as follows: derived NLR 2.02 (range: 0.87–7.96), NLR 3.04 (range: 0.46–11.24), PLR 189.7 (range: 52.72–779.62), and LMR 2.41 (range: 0.80–8.13). The median hemoglobin level was 12.6 g/dL (range: 7.5–16.1). The median age at diagnosis was 56 years (range: 32–83). The findings are summarized in Table 3.

No statistically significant relationship was observed between the mean recurrence-free survival (disease-free interval) and pathological differentiation, TRG, or tumor response to treatment (*p* > 0.05). The corresponding findings are summarized in Table 4.

No statistically significant relationship was observed between TRG and differentiation grades and baseline laboratory parameters, including hemoglobin, neutrophil count, lymphocyte count, D-NLR, NLR, and PLR (*p* > 0.05). The corresponding findings are summarized in Table 5 and Table 6.

When recurrence time was used as a reference, TRG was analyzed using Spearman’s Rho correlation with preoperative inflammatory markers. No statistically significant relationship was observed between complete response, good response, or moderate response and baseline laboratory parameters, including hemoglobin, neutrophil count, lymphocyte count, D-NLR, NLR, and PLR (*p* > 0.05). However, while poor response was not significantly associated with hemoglobin, neutrophil, and lymphocyte levels, a weak but statistically significant negative correlation was detected with D-NLR, NLR, and PLR (*p* < 0.05 *). The corresponding findings are summarized in Table 7.

When recurrence time was used as a reference, tumor differentiation grade, including moderate and poor differentiation, was analyzed using Spearman’s Rho and Pearson correlation tests with preoperative laboratory parameters. No statistically significant relationship was observed between hemoglobin, neutrophil count, lymphocyte count, D-NLR, NLR, and PLR (*p* > 0.05). However, while well-differentiated tumors showed no significant correlation with hemoglobin, neutrophil count, lymphocyte count, and NLR, a weak but statistically significant negative correlation was detected with D-NLR and PLR (*p* < 0.05 *). The corresponding findings are summarized in Table 8.

No statistically significant difference was observed between the pCR to treatment and baseline laboratory parameters, including hemoglobin, neutrophil count, lymphocyte count, D-NLR, NLR, PLR, and LMR (*p* > 0.05). The corresponding findings are summarized in Table 9.

To assess whether pre-chemotherapy serum hemoglobin, neutrophil, lymphocyte, D-NLR, NLR, PLR, and LMR values could predict tumor differentiation response, patients were categorized into two groups based on post-chemotherapy tumor response: those with a good response and those with a moderate or poor response. Laboratory values were compared between these groups by calculating optimal cutoff values with appropriate specificity and sensitivity. Similarly, the specificity and sensitivity of laboratory parameters were also evaluated in relation to TRG and recurrence-free survival.

To determine inflammatory markers, prognosis, and treatment response in patients with locally advanced RC, ROC analysis was performed for various laboratory parameters. The ROC curves and their corresponding Area Under the Curve (AUC) values for tumor differentiation grade, TRG, and recurrence status were analyzed.

For tumor differentiation, none of the evaluated inflammatory markers, including hemoglobin, neutrophil count, lymphocyte count, derived NLR, NLR, PLR, and LMR, showed significant discriminatory ability. The AUC values for these markers ranged from 0.447 to 0.612, with *p*-values exceeding the threshold for statistical significance (Figure 1, Table 10).

Similarly, in the analysis of TRG, the predictive performance of inflammatory markers remained limited. The AUC values varied between 0.489 and 0.622, and no statistically significant associations were observed (*p* > 0.05) (Figure 2, Table 11).

Regarding recurrence-free survival, the ROC analysis demonstrated AUC values ranging from 0.525 to 0.643. Although hemoglobin exhibited the highest AUC value (0.643 ± 0.080, *p* = 0.062), it did not reach statistical significance. Other inflammatory markers also failed to show significant prognostic value (Figure 3, Table 12).

## 4. Discussion

Curative treatment for RC is primarily based on surgery, with neoadjuvant CRT increasingly utilized for locally advanced RC to improve local control and achieve better resection outcomes. Neoadjuvant CRT has been shown to reduce tumor size, enhance the feasibility of sphincter-preserving surgery, and lower local recurrence rates [6,16,17,18]. However, the response to neoadjuvant treatment varies among patients, necessitating the identification of reliable predictive biomarkers.

Several studies have investigated the prognostic value of inflammatory markers, including the NLR, PLR, LMR, and individual blood parameters in predicting treatment response and survival outcomes. Shen et al. found that pre-treatment NLR, lymphocyte, white blood cell, and platelet levels did not significantly predict pCR, although a high NLR was associated with poor prognosis and tumor recurrence [19]. Similarly, Carruthers et al. demonstrated that elevated NLR, white blood cell count, and platelet levels correlated with shorter survival and increased recurrence risk, whereas other markers such as CEA, albumin, and lymphocyte count did not show significant associations [20].

In contrast, Kitayama et al. reported that lymphocyte levels were positively associated with improved prognosis and response to neoadjuvant therapy, while neutrophil levels were higher in patients with poor response [21]. Krauthamer et al. also concluded that a lower NLR was linked to better tumor response following neoadjuvant therapy [22]. Additionally, Kim et al. found that lower pre-treatment NLR, neutrophil count, and white blood cell count were associated with a favorable response to therapy, while lymphocyte, platelet, and albumin levels showed no predictive value [23]. However, Nagasaki et al. and Heo et al. reported that NLR was not a significant predictor of pCR, although higher NLR values were correlated with worse survival and recurrence [24,25].

Meta-analyses have also provided conflicting results. Zhang et al. analyzed data from 23 studies, including 11,762 patients, and concluded that high NLR, PLR, and platelet counts were indicators of poor prognosis in CRC, with NLR specifically being linked to increased recurrence [26]. However, our study did not find a statistically significant association between these inflammatory markers and tumor differentiation, regression grade, or pCR.

Regarding treatment response and recurrence, studies by Kim et al. and Ryan et al. demonstrated that patients achieving pCR had significantly lower recurrence rates [14,27]. Maas et al. conducted a meta-analysis of 3105 patients, reporting a pCR rate of 8–24% and a 5-year DFS rate of 83.3% [9]. Similarly, Kalady et al. and Lobato et al. found pCR rates of approximately 24%, with poor tumor differentiation being observed in 71.7% of cases [8,28].

Martin et al. conducted a meta-analysis including 16 studies with a total of 3363 patients, with a mean age of 60 years, 65% male and 35% female. They reported a pCR rate of 37.5% (1263 patients) and a 5-year recurrence-free survival rate of 87% [29].

Takeo et al. analyzed 67 patients with locally advanced RC who received neoadjuvant CRT, with an age range of 32–79 years (median: 63), 64% male and 36% female. They found a pCR rate of 34.7% (24 patients) and a median OS of 54.4 months [30].

Our study did not identify a statistically significant association between inflammatory markers and tumor differentiation, regression grade, or pCR in locally advanced RC. While some studies have suggested that markers such as NLR, PLR, and LMR may have prognostic significance [19,20,21,22,23,24,25,26], our findings align with those of Shen et al. and Nagasaki et al., who also reported no predictive value for these inflammatory parameters in response to neoadjuvant CRT [19,24].

To the best of our knowledge, few studies have directly examined the correlation between inflammatory markers and tumor regression grade (TRG) following neoadjuvant therapy. In our cohort, we observed a significant inverse association between NLR, dNLR, PLR, and TRG scores, suggesting that a heightened inflammatory state may hinder tumor regression. While this relationship warrants further investigation, our findings are in line with the broader literature linking elevated systemic inflammation to impaired treatment response and poor oncologic outcomes.

The variability in reported findings across different studies suggests that inflammatory markers alone may not serve as reliable predictors of treatment response or survival outcomes. Differences in patient populations, treatment protocols, and cutoff values for biomarker stratification may contribute to the inconsistencies observed in the literature. Some meta-analyses, including those by Zhang et al. and Martin et al., have suggested that elevated NLR and PLR correlate with poorer prognosis and recurrence risk [26,29]. However, our study, in line with other investigations, did not confirm these associations.

Pathological complete response rates in our study were comparable to those reported in previous studies, such as the meta-analysis by Maas et al. [9]. While some studies, including those by Kalady et al. and Takeo et al., have reported slightly higher pCR rates [28,30], others, such as Kim et al., observed lower response rates [27]. The observed variability in response to neoadjuvant therapy underscores the complexity of predicting treatment outcomes in RC and highlights the need for more comprehensive predictive models incorporating molecular and histopathological factors.

## 5. Conclusions

Neoadjuvant chemoradiotherapy has become a standard approach in the treatment of locally advanced RC, improving tumor resectability and reducing local recurrence rates. However, predicting treatment response remains a challenge, and various inflammatory markers have been proposed as potential prognostic indicators.

In our retrospective study with a limited sample size, we did not find a statistically significant association between inflammatory markers and tumor differentiation, regression grade, or pCR. While some previous studies have suggested a prognostic role for NLR, PLR, and LMR, our findings align with studies that reported no predictive value for these markers in assessing neoadjuvant treatment response. The discrepancy between our findings and those of previous studies that reported stronger associations between inflammatory markers and treatment response may be explained by several factors. Differences in study design, patient selection criteria, timing and methodology of blood sample collection, variation in the definition of treatment response (e.g., pCR vs. TRG), and heterogeneity in neoadjuvant treatment protocols could all contribute to this variability. Furthermore, inconsistencies in the cutoff values used for inflammatory markers across studies may have affected statistical significance and comparability.

The retrospective design and relatively small cohort of our study limit the generalizability of our findings. Additionally, variations in cutoff values for inflammatory markers across different studies make it difficult to establish a standardized predictive model. Future prospective studies with larger patient cohorts and standardized methodologies are needed to determine the true clinical relevance of these inflammatory markers. A more comprehensive approach incorporating molecular, genetic, and histopathological factors may provide better insights into treatment response prediction in RC. Integrating inflammatory biomarkers with molecular profiles and radiologic response parameters has the potential to improve the accuracy of predictive models for treatment response in rectal cancer.

One of the limitations of our study is that we did not perform a multivariate analysis to control for other factors that might affect the results. Variables such as age, tumor stage, and the type of neoadjuvant treatment (chemoradiotherapy versus radiotherapy alone) could potentially influence both the levels of inflammatory markers and treatment outcomes. Since our study was retrospective and included a relatively small number of patients, we were unable to adjust for these possible confounding factors in the analysis. Therefore, our results should be interpreted with this limitation in mind. Larger and prospective studies are needed to confirm our findings and to better understand the role of these biomarkers after accounting for such factors.

## Figures and Tables

**Figure 1 medicina-61-00865-f001:**
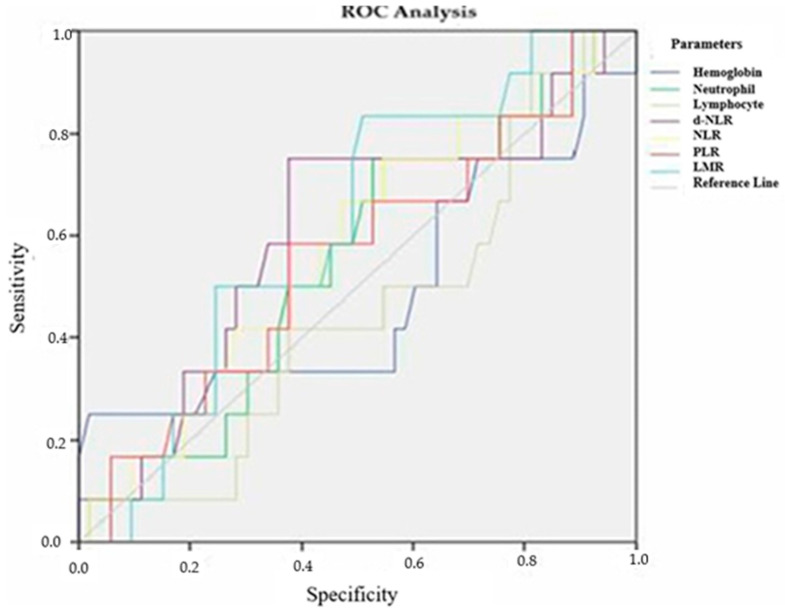
ROC analysis of laboratory parameters according to tumor differentiation grade.

**Figure 2 medicina-61-00865-f002:**
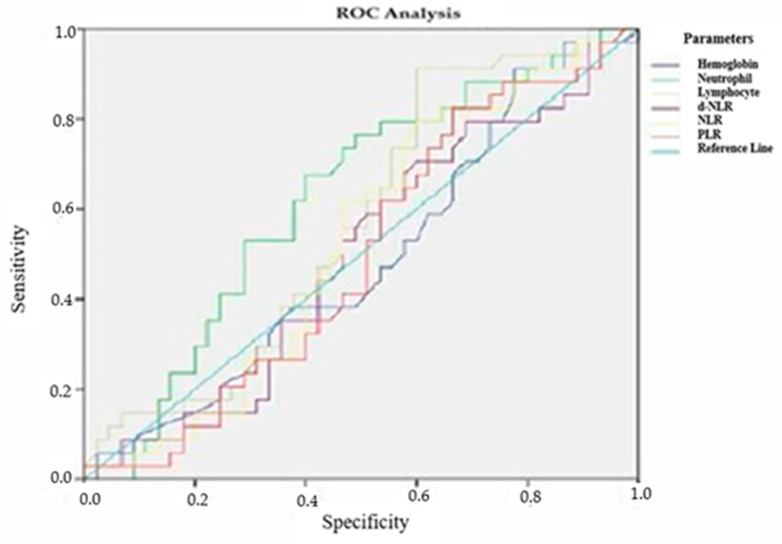
ROC analysis of laboratory parameters according to tumor regression grade.

**Figure 3 medicina-61-00865-f003:**
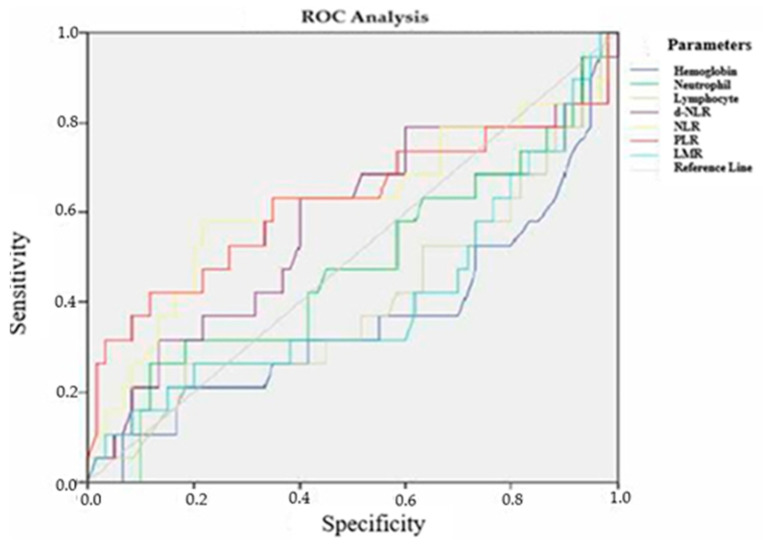
ROC analysis of laboratory parameters according to recurrence time.

**Table 1 medicina-61-00865-t001:** Distribution of tumor staging based on TNM classification (frequency and percentage).

Stage	Percentage (*n* = 79)
I	11.4% (9)
2A	11.4% (9)
2B	5.1% (4)
2C	3.8% (3)
3A	24.1% (19)
3B	26.6% (21)
3C	10.1% (8)
4A	5.1% (4)
4B	2.5% (2)

**Table 2 medicina-61-00865-t002:** Distribution of tumor invasion status and radiotherapy type by count and percentage.

Parameters	Percentage (*n* = 79)
Anal sphincter distance (cm)	
1–4	40.5% (32)
5–10	48.1% (38)
11–15	11.4% (9)
LVI	
No	79.7% (63)
Yes	20.3% (16)
PNI	
No	78.5% (62)
Yes	21.5% (17)
Circumferential resection margin invasion	
No	100% (79)
Yes	0% (0)
Radiotherapy type	
RT	27.8% (22)
CRT	72.2% (57)

LVI: lymphovascular invasion, PNI: perineural invasion, RT: radiotherapy, and CRT: chemoradiotherapy.

**Table 3 medicina-61-00865-t003:** Median age and laboratory values at the time of diagnosis.

Parameters	Median (Min–Max)
Lymphocyte	1390 (320–4090)
Derived NLR	2.02 (0.87–7.96)
NLR	3.04 (0.46–11.24)
PLR	189.7 (52.72–779.62)
LMR	2.41 (0.80–8.13)
Hemoglobin	12.6 (7.5–16.10)
Neutrophil	4520 (1940–12,050)
Age	56 (32–83)

NLR: neutrophil-to-lymphocyte ratio, PLR: platelet-to-lymphocyte ratio, and LMR: lymphocyte-to-monocyte ratio.

**Table 4 medicina-61-00865-t004:** Recurrence-free survival based on tumor differentiation grade.

Parameters	Time to Relapse (Months) Median (Min–Max)	*p*
Differentiation *n* = 65		0.197
G1 = 53	16 (6–50)	
G2 = 4	11 (5–14)	
G3 = 8	27.5 (6–43)	
Tumor Regression *n* = 79		0.776
G0 = 10	12.5 (6–50)	
G1 = 24	15.5 (5–47)	
G2 = 29	15 (6–44)	
G3 = 16	18 (7–50)	
Response *n* = 79		0.685
Complete Response = 10	12.5 (6–50)	
Non-Complete = 69	15 (5–50)	

**Table 5 medicina-61-00865-t005:** Median laboratory values based on tumor regression grade.

Parameters *n* = 79 Tumor Regression Grade	G0 Median (Min–Max)	G1 Median (Min–Max)	G2 Median (Min–Max)	G3 Median (Min–Max)	*p*
Hemoglobin	13.05 (11.1–15.7)	12.55 (7.5–16.1)	12.6 (9–16.5)	12.65 (9.6–16)	*p* > 0.05
Neutrophil	4760 (2520–6330)	5270 (2600–7860)	3990 (1940–8370)	4465 (2550–12,050)	0.210
Lymphocyte	1315 (540–3200)	1505 (600–4090)	1170 (320–3970)	1770 (610–2830)	0.533
Derived-NLR	2.04 (1.32–4.66)	2.02 (0.87–4.16)	2.00 (0.87–4.59)	1.815 (1.07–7.96)	0.941
NLR	2.815 (1.91–11.24)	3.42 (1.1–7.21)	2.86 (0.46–10.09)	2.74 (1.39–10.68)	0.921
PLR	192.27 (12.18–779.62)	183.25 (62.97–347)	222.00 (57.72–603)	142.66 (66.00–491.80)	0.957

NLR: neutrophil-to-lymphocyte ratio, PLR: platelet-to-lymphocyte ratio.

**Table 6 medicina-61-00865-t006:** Median laboratory values based on tumor differentiation grade.

Parameters *n* = 65 Differentiation	G1 Median (Min–Max)	G2 Median (Min–Max)	G3 Median (Min–Max)	*p*
Hemoglobin	12.55 (7.5–16.1)	12.6 (9–16.5)	12.65 (9.6–16)	*p* > 0.05
Neutrophil	4520 (1940–12,050)	4895 (3260–7150)	5055 (2560–8440)	0.891
Lymphocyte	1390 (320–4090)	1805 (900–2100)	830 (600–3350)	0.400
Derived-NLR	1.96 (0.87–4.66)	1.85 (1.18–3.25)	2.81 (1.33–7.96)	0.197
NLR	2.99 (0.46–11.24)	2.63 (1.55–5.02)	4.85 (1.74–10.68)	0.278
PLR	189.51 (52.72–779.62)	176.36 (95.85–326.66)	256.00 (91.64–468.30)	0.608

NLR: neutrophil-to-lymphocyte ratio, PLR: platelet-to-lymphocyte ratio.

**Table 7 medicina-61-00865-t007:** Relationship between tumor regression grade and laboratory parameters.

Recurrence Time	Hemoglobin	Neutrophil	Lymphocyte	D-NLR	NLR	PLR
All Groups *n* = 79 Tumor Regression	Rho: 0.132 *p*: 0.246	Rho: −0.130 *p*: 0.253	Rho: 0.045 *p*: 0.694	Rho: −0.129 *p*: 0.257	Rho: −0.089 *p*: 0.438	Rho: −0.175 *p*: 0.112
G0 *n* = 10	Rho: 0.088 *p*: 0.808	Rho: 0.085 *p*: 0.815	Rho: 0.444 *p*: 0.199	Rho: −0.122 *p*: 0.738	Rho: −0.366 *p*: 0.298	Rho: −0.292 *p*: 0.413
G1 *n* = 24	Rho: 0.337 *p*: 0.107	Rho: −0.213 *p*: 0.317	Rho: −0.214 *p*: 0.315	Rho: 0.019 *p*: 0.929	Rho: 0.148 *p*: 0.489	Rho: −0.009 *p*: 0.965
G2 *n* = 29	Rho: −0.012 *p*: 0.950	Rho: −0.057 *p*: 0.769	Rho: 0.021 *p*: 0.912	Rho: −0.043 *p*: 0.825	Rho: 0.082 *p*: 0.672	Rho: −0.043 *p*: 0.824
G3 *n* = 16	Rho: 0.029 *p*: 0.914	Rho: −0.355 *p*: 0.177	Rho: 0.243 *p*: 0.364	Rho: −0.514 *p*: 0.042 *	Rho: −0.514 *p*: 0.042 *	Rho: −0.522 *p*: 0.038 *

NLR: neutrophil-to-lymphocyte ratio, PLR: platelet-to-lymphocyte ratio, * “*p* < 0.05”.

**Table 8 medicina-61-00865-t008:** Relationship between tumor differentiation grade and laboratory parameters.

Recurrence Time	Hemoglobin	Neutrophil	Lymphocyte	D-NLR	NLR	PLR
All Groups *n* = 65 Differentiation	Rho: 0.148 *p:* 0.239	Rho: −0.178 *p:* 0.157	Rho: 0.060 *p:* 0.635	Rho: −0.227 *p:* 0.068	Rho: −0.126 *p:* 0.318	Rho: −0.205 *p:* 0.102
G1 *n* = 53	Rho: 0.337 *p:* 0.107	Rho: −0.145 *p:* 0.300	Rho: 0.139 *p:* 0.320	Rho: −0.343 *p:* 0.012 *	Rho: −0.208 *p:* 0.135	Rho: −0.279 *p:* 0.043 *
G2 *n* = 4	Rho: −0.012 *p:* 0.950	Rho: −0.316 *p:* 0.684	Rho: −0.738 *p:* 0.262	Rho: 0.738 *p:* 0.262	Rho: 0.738 *p:* 0.262	Rho: 0.949 *p:* 0.051
G3 *n* = 8	Rho: 0.029 *p:* 0.914	Rho: −0.180 *p:* 0.670	Rho: −0.263 *p:* 0.528	Rho: −0.024 *p* = 0.955	Rho: 0.060 *p* = 0.888	Rho: −0.096 *p* = 0.821

NLR: neutrophil-to-lymphocyte ratio, PLR: platelet-to-lymphocyte ratio, * “*p* < 0.05”.

**Table 9 medicina-61-00865-t009:** Evaluation of laboratory parameters based on tumor complete response status.

Parameters PCR *n* = 79	Median (Min–Max) Non-CR *n* = 69	Median (Min–Max) Complete Response *n* = 10	*p*
Neutrophil	4520 (1940–12,050)	4760 (2520–6330)	0.889
Lymphocyte	1390 (320–4090)	1315 (540–3200)	0.912
D-NLR	2.01 (0.87–7.96)	2.04 (1.32–4.66)	0.819
NLR	3.07 (0.46–10.68)	2.81 (1.91–11.24)	0.831
PLR	189.51 (52.72–603.00)	192.27 (112.18–779.62)	0.930
LMR	2.29 (0.80–6.37)	2.65 (0.83–8.13)	0.327

NLR: neutrophil-to-lymphocyte ratio, PLR: platelet-to-lymphocyte ratio, and LMR: lymphocyte-to-monocyte ratio.

**Table 10 medicina-61-00865-t010:** ROC analysis and sensitivity values of laboratory parameters according to tumor differentiation grade.

Parameters Group 1 (G1) Group 2 (G2 and G3) Differentiation *n* = 65	Cutoff Value	Sensitivity	Specificity	Area Under the Curve (AUC)	*p*
Hemoglobin	11.30	66.7%	35.8%	0.484 + −0.107	0.866
Neutrophil	4420	75.0%	47.2%	0.549 + −0.088	0.600
Lymphocyte	1220	50.0%	45.3%	0.447 + −0.087	0.565
D-NLR	2.06	75.0%	62.3%	0.605 + −0.095	0.257
NLR	2.76	75.0%	45.3%	0.573 + −0.091	0.432
PLR	170.6	66.7%	47.2%	0.554 + −0.093	0.560
LMR	2.13	83.3%	49.1%	0.612 + −0.080	0.227

NLR: neutrophil-to-lymphocyte ratio, PLR: platelet-to-lymphocyte ratio, and LMR: lymphocyte-to-monocyte ratio.

**Table 11 medicina-61-00865-t011:** Specificity and sensitivity values of laboratory parameters according to tumor regression.

Parameters Group 1 (G0, G1) Group 2 (G2, G3) Tumor Regression	Cutoff Value	Sensitivity	Specificity	Area Under the Curve (AUC)	*p*
Hemoglobin	12.35	50%	42.2%	0.489 + −0.066	0.866
Neutrophil	4270	73.5%	53.3%	0.622 + −0.064	0.065
Lymphocyte	1005	73.5%	44.4%	0.572 + −0.065	0.276
D-NLR	1.65	70.6%	40%	0.495 + −0.066	0.941
NLR	2.35	79.4%	40%	0.527 + −0.065	0.681
PLR	162.6	61.8%	46.7%	0.493 + −0.065	0.913
LMR	12.05	59.4%	38.1%	0.523 + −0.068	0.731

NLR: neutrophil-to-lymphocyte ratio, PLR: platelet-to-lymphocyte ratio, and LMR: lymphocyte-to-monocyte ratio.

**Table 12 medicina-61-00865-t012:** Specificity and sensitivity values of laboratory parameters according to recurrence time.

Parameters Recurrence Time	Cutoff Value	Sensitivity	Specificity	Area Under the Curve (AUC)	*p*
Hemoglobin	13.55	52.6%	26.7%	0.643 + −0.080	0.062
Neutrophil	5175	52.6%	41.7%	0.525 + −0.082	0.748
Lymphocyte	1665	52.6%	36.7%	0.597 + −0.081	0.205
D-NLR	1.98	63.2%	60%	0.572 + −0.082	0.347
NLR	2.73	63.2%	65%	0.618 + −0.085	0.121
PLR	162.6	63.2%	65%	0.621 + −0.088	0.113
LMR	3.55	63.2%	23.3%	0.593 + −0.082	0.224

NLR: neutrophil-to-lymphocyte ratio, PLR: platelet-to-lymphocyte ratio, and LMR: lymphocyte-to-monocyte ratio.

## Data Availability

While the datasets analyzed in this study are not publicly accessible, they can be obtained from the corresponding author upon reasonable request.

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
