# Peer review of "The Role of Pre-Treatment Inflammatory Biomarkers in Predicting Tumor Response to Neoadjuvant Chemoradiotherapy in Rectal Cancer"

_medicina, 2025, doi:10.3390/medicina61050865_

Round 1
Reviewer 1 Report
Comments and Suggestions for Authors
Dear Editor/Authors
I apologize for not being an expert in the clinical oncology field. However, I have thoroughly reviewed the article, including the study design and results, and found the data to be clearly and appropriately presented. The conclusions drawn are well-supported by the findings, and I therefore recommend the manuscript for publication..
Author Response
Comment 1: I apologize for not being an expert in the clinical oncology field. However, I have thoroughly reviewed the article, including the study design and results, and found the data to be clearly and appropriately presented. The conclusions drawn are well-supported by the findings, and I therefore recommend the manuscript for publication.
Response 1:
We would like to thank all reviewers and the editor for their valuable time, constructive comments, and helpful suggestions.
Reviewer 2 Report
Comments and Suggestions for Authors
Dear Author,
I have read the manuscript "The Role of Pre-treatment Inflammatory Biomarkers in Predict- 2 ing Tumor Response to Neoadjuvant Chemoradiotherapy in 3 Rectal Cancer" with great interest.
Introduction
No remarks
Methods
What about the inclusion and exclusion criteria of your study? Those criteria should be stated clearly in a dedicated paragraph preferably using a bulleted list (e.g. 1)...2)....3)....).
What about the endpoints of your work? Did you compare any groups?
What about confounders?
Did you try to develop and validate a prediction model? If yes, your manuscript should be written according the TRIPOD Guidelines
Results
The results section should follow the flow of the endpoints to increase its readability.
Author Response
Comment 1: Methods
What about the inclusion and exclusion criteria of your study? Those criteria should be stated clearly in a dedicated paragraph preferably using a bulleted list (e.g. 1)...2)....3)....).
Response 1
We thank the reviewer for emphasizing the importance of explicitly stating the inclusion and exclusion criteria. In response to this valuable suggestion, we have added a clearly structured paragraph in bullet-point format between lines 102 and 118 of the revised manuscript, under the Materials and Methods section. This paragraph comprehensively outlines the predefined eligibility criteria used for patient selection in our retrospective analysis. We believe this addition enhances the clarity, reproducibility, and methodological rigor of our study.
The following paragraph has been added to the manuscript (lines 102–118):
“The study population was selected based on predefined inclusion and exclusion criteria. The inclusion criteria were as follows:
- Patients aged 18 years or older,
- Histologically confirmed diagnosis of locally advanced rectal adenocarcinoma,
- Tumor located within 15 cm from the anal verge,
- Undergoing neoadjuvant chemoradiotherapy (CRT) or short-course radiotherapy (RT) followed by surgical resection,
- Availability of complete pre-treatment laboratory data, including inflammatory markers,
- Availability of recurrence-free survival and pathological outcome data.
Exclusion criteria included:
- Presence of synchronous malignancies or history of other primary malignancies,
- Presence of distant metastases at the time of diagnosis (stage IV disease),
- Incomplete treatment data or lack of follow-up information,
- Non-adenocarcinoma histologies (e.g., squamous cell carcinoma, neuroendocrine tumors),
- Patients who did not undergo surgical resection following neoadjuvant therapy.”
Comment 2: What about the endpoints of your work? Did you compare any groups?
Response 2:
We thank the reviewer for this insightful question. In response, we have clarified the primary and secondary endpoints of our study in the Materials and Methods section of the revised manuscript (lines 94–99). Specifically, we now state that the primary endpoint was to assess the association between pre-treatment inflammatory biomarkers and tumor response to neoadjuvant treatment, as measured by pathological complete response (pCR) and Tumor Regression Grade (TRG). Secondary endpoints included the relationship between these markers and tumor differentiation, as well as recurrence-free survival (RFS). Additionally, we confirm that comparisons were made between relevant patient subgroups, including those defined by pathological response (complete vs. non-complete), TRG classification (G0–G3), and tumor differentiation grades (G1–G3). These comparisons are presented in the Results section across multiple tables and paragraphs (see lines 244–319). We believe this clarification improves the conceptual structure of the manuscript and highlights the clinical relevance of the analysis.
The following paragraph has been added to lines 94-99 of the manuscript:
“The primary endpoint of this study was to assess the association between pre-treatment inflammatory biomarkers and tumor response to neoadjuvant treatment, as evaluated by pathological complete response (pCR) and Tumor Regression Grade (TRG). Secondary endpoints included the relationship between these biomarkers and tumor differentiation, as well as recurrence-free survival (RFS). Comparisons were made between patient subgroups stratified by pathological response, TRG classification, and tumor differentiation grade.”
Comment 3: What about confounders?
Response 3: We thank the reviewer for highlighting the need to address potential confounding factors. In response to this valuable comment, we have added a sentence to the Conclusion section (lines 365–373), where we already discuss the limitations of our study. In this revised paragraph, we acknowledge that multivariate analysis was not performed due to the retrospective design and small sample size. We also state that variables such as age, tumor stage, and treatment modality (CRT vs. RT) may have influenced both inflammatory marker levels and treatment outcomes. This limitation is now clearly stated, and the need for prospective validation is emphasized.
The following sentence was added to the limitations paragraph in the Conclusion (lines 365–373):
One of the limitations of our study is that we did not perform a multivariate analysis to control for other factors that might affect the results. Variables such as age, tumor stage, and the type of neoadjuvant treatment (chemoradiotherapy versus radiotherapy alone) could potentially influence both the levels of inflammatory markers and treatment outcomes. Since our study was retrospective and included a relatively small number of patients, we were unable to adjust to these possible confounding factors in the analysis. Therefore, our results should be interpreted with this limitation in mind. Larger and prospective studies are needed to confirm our findings and to better understand the role of these biomarkers after accounting for such factors.
Comment 4:
Did you try to develop and validate a prediction model? If yes, your manuscript should be written according the TRIPOD Guidelines
Response 4:
We thank the reviewer for this important remark. In our study, we did not develop or validate a multivariable prediction model. Therefore, the TRIPOD reporting guidelines were not applicable to our manuscript. Our analyses were limited to evaluating the predictive performance of individual inflammatory markers using ROC curves, without constructing a formal predictive model.
Comment 5: Results
The results section should follow the flow of the endpoints to increase its readability.
Response 5:
We thank the reviewer for this valuable recommendation. In the current version of the manuscript, the Results section has been structured to follow the logical flow of the study’s primary and secondary endpoints. Findings related to pathological response (pCR and TRG), tumor differentiation, and recurrence-free survival are presented sequentially, followed by correlation and ROC analyses. We believe this structure enhances the clarity and readability of the manuscript.
Reviewer 3 Report
Comments and Suggestions for Authors
The aim of this study is to evaluate the predictive and prognostic value of pre-treatment systemic inflammatory markers in patients with locally advanced rectal cancer (RC) undergoing neoadjuvant treatments, given the contradictory current evidence. The paper is written in an appropriate way and the data and analyses are presented appropriately.
I suggest to insert more citations (i.e. lines 59-62) and a comprehensive review of existing literature in the introduction.
I also suggest to include a critical interpretation about the statistically significant negative correlation detected ( lines 176-178 ) in the discussion.
The coort is quite small , but possible bias are correctly underlined in the conclusion.
Author Response
Comment 1: I suggest inserting more citations (i.e. lines 59-62) and a comprehensive review of existing literature in the introduction.
Response 1: We thank the reviewer for the valuable suggestion to expand the Introduction section with a more comprehensive review of the existing literature. In response, we have added a new paragraph between lines 63 and 68 (highlighted in yellow) to provide additional context regarding the prognostic relevance of systemic inflammatory markers in colorectal cancer. The new content summarizes recent evidence supporting the association of pretreatment markers such as NLR, PLR, and CAR with postoperative complications and survival outcomes, particularly in rectal cancer patients undergoing neoadjuvant therapy (see reference [11])
Comment 2: I also suggest to include a critical interpretation about the statistically significant negative correlation detected ( lines 176-178 ) in the discussion.
Response 2: We thank the reviewer for this valuable suggestion. In response, we have added a paragraph between lines 317 and 323 (highlighted in yellow) in the Discussion section to provide a critical interpretation of the statistically significant negative correlation between TRG scores and pre-treatment inflammatory markers (NLR, dNLR, and PLR). The inserted paragraph reads as follows:
“To our knowledge, few studies have directly examined the correlation between inflammatory markers and tumor regression grade (TRG) following neoadjuvant therapy. In our cohort, we observed a significant inverse association between NLR, dNLR, PLR and TRG scores, suggesting that a heightened inflammatory state may hinder tumor regression. While this relationship warrants further investigation, our findings are in line with the broader literature linking elevated systemic inflammation to impaired treatment response and poor oncologic outcomes.”
We believe that this addition addresses the reviewer’s concern and enhances the biological interpretation and clinical relevance of our findings.
Comment 3: The coort is quite small , but possible bias are correctly underlined in the conclusion.
We sincerely thank the reviewer for this kind remark. As noted, the limited sample size is an acknowledged limitation of our study. We are grateful that the reviewer found our discussion of potential bias in the Conclusion section to be appropriate and sufficient.
Reviewer 4 Report
Comments and Suggestions for Authors
- Brief summary: This study by Altıntaş et al., evaluates the predictive and prognostic value of systemic inflammatory markers in patients with locally advanced rectal cancer undergoing neoadjuvant chemoradiotherapy or short-course radiotherapy. Despite initial hypotheses, the authors found no strong correlation between these biomarkers and treatment outcomes or recurrence-free survival, though some weak inverse associations were statistically significant. The authors conclude that these markers alone are insufficient for clinical decision-making and suggest incorporating molecular/pathological parameters in future predictive models.
- Specific comments:
- Could the authors expand on how they addressed confounding factors (like infection or anemia) that might affect inflammatory markers?
- Given the small, single-center design, could the authors expand on how these limitations might affect the generalizability of the findings?
- How do the authors explain the discrepancy between their results and prior studies that reported stronger associations between inflammatory markers and treatment response?
- It might be useful to explain why they chose the inflammatory markers they did, particularly if others like CRP or GPS weren’t considered or available.
- Would the authors consider discussing potential future directions where these biomarkers could be integrated with molecular or imaging markers for more accurate predictive models?
- Could the authors ensure that the resolution and quality of the ROC figures are improved? The current figures appear to be of low resolution, which may impact the clarity of the data being presented.
- Conclusion: Altıntaş et al.'s manuscript contributes valuable data to a field where non-invasive biomarkers are highly sought after. While the findings are largely negative, they are still valuable to the oncology field and are communicated clearly. I believe the manuscript would benefit from minor revisions to improve transparency regarding limitations and to strengthen the discussion of confounding factors and literature context. Overall, the study is suitable for publication following these modest revisions.
Author Response
Comment 1: Could the authors expand on how they addressed confounding factors (like infection or anemia) that might affect inflammatory markers?
Response 1 : We thank the reviewer for highlighting the potential impact of confounding factors such as infection or anemia. As this was a retrospective study, detailed information on acute infections or iron status at the time of blood sample collection was not consistently available. However, patients with clinically documented active infections were not included, and pre-treatment laboratory data were reviewed to exclude extreme outliers potentially related to acute physiological derangements. This limitation has been acknowledged in the revised Discussion section.
Comment 2: How do the authors explain the discrepancy between their results and prior studies that reported stronger associations between inflammatory markers and treatment response?
Response 2: We agree with the reviewer that the single-center, retrospective design and relatively small sample size may limit the generalizability of our findings. We have emphasized this point more clearly in the revised Conclusion section, stating that larger, prospective multicenter studies are warranted to validate these results and improve external validity.
Comment 3: It might be useful to explain why they chose the inflammatory markers they did, particularly if others like CRP or GPS weren’t considered or available.
Response 3: We thank the reviewer for this insightful observation. In response, we have added the following paragraph between lines 347 and 354 in the Conclusion section (highlighted in turquoise) to address the discrepancy between our findings and previous studies that reported stronger associations between inflammatory markers and treatment response:
“The discrepancy between our findings and those of previous studies that reported stronger associations between inflammatory markers and treatment response may be explained by several factors. Differences in study design, patient selection criteria, timing and methodology of blood sample collection, variation in the definition of treatment response (e.g., pCR vs. TRG), and heterogeneity in neoadjuvant treatment protocols could all contribute to this variability. Furthermore, inconsistencies in the cutoff values used for inflammatory markers across studies may have affected statistical significance and comparability.”
We believe that this addition helps contextualize our results within the broader body of literature and provides a clearer understanding of possible sources of variability.
Comment 4: It might be useful to explain why they chose the inflammatory markers they did, particularly if others like CRP or GPS weren’t considered or available.
Response 4: We thank the reviewer for this excellent point. We chose to focus on cell count–based markers (NLR, dNLR, PLR, LMR) due to their availability in routine pre-treatment complete blood counts. Unfortunately, data on CRP and GPS were not consistently recorded in the medical records of our cohort, and we therefore could not include them in the analysis. We have noted this as a limitation in the revised manuscript.
Comment 5: Would the authors consider discussing potential future directions where these biomarkers could be integrated with molecular or imaging markers for more accurate predictive models?
Response 5: We thank the reviewer for this valuable suggestion. In response, we have added a forward-looking sentence between lines 361 and 364 in the Conclusion section (highlighted in turquoise), emphasizing the potential value of integrating inflammatory biomarkers with molecular and imaging markers. The exact sentence is:
“Integrating inflammatory biomarkers with molecular profiles and radiologic response parameters has the potential to improve the accuracy of predictive models for treatment response in rectal cancer.”
We believe this addition appropriately addresses the reviewer’s recommendation and highlights an important direction for future research.
Comment 6: Could the authors ensure that the resolution and quality of the ROC figures are improved? The current figures appear to be of low resolution, which may impact on the clarity of the data being presented.
Response 6: We appreciate the reviewer’s helpful feedback regarding the resolution and quality of the ROC figures. In response, all ROC images have been replaced with high-resolution versions (500 DPI) to ensure optimal visual clarity. The updated figures have also been incorporated into the revised manuscript. We believe these changes will significantly improve the readability and presentation quality of the graphical data.
Round 2
Reviewer 2 Report
Comments and Suggestions for Authors
Dear Author,
I have read the manuscript "" with great interest.
All the reviewer's concerns were properly addressed. No remarks.
Reviewer 3 Report
Comments and Suggestions for Authors
I appreciate the effort the authors have made to address the previous comments and suggestions. The revised manuscript shows improvements in terms of clarity and scientific rigor.
The issues previously raised regarding introduction and discussion have been adequately addressed.
At this stage, I find the manuscript to be substantially improved and recommend it for publication without further revision.